# Making silver a stronger n-dopant than cesium via in situ coordination reaction for organic electronics

Zhengyang Bin[1,2], Guifang Dong[1], Pengcheng Wei[1], Ziyang Liu[1], Dongdong Zhang[1], Rongchuan Su[2], Yong Qiu[1] & Lian Duan[1,3]

N-doping is an effective way to increase the electron conductivity of organic semiconductors and achieve ohmic cathode contacts in organic electronics. To avoid the use of difficult-to-handle highly reactive n-dopants, air-stable precursors are widely used nowadays, which could decompose to release reactive species in a subtractive way though always with unwanted and even harmful byproducts during processing. Here, we show that air-stable metals, such as copper, silver and gold, could release free electrons readily in the presence of chelating ligands, as the irreversible coordination reaction between metal ions and the ligands would push the equilibrium between metals and metal ions to the forward direction. By using a well-designed multi-functional electron transport material with a strong nucleophilic quality, 4,7-dimethoxy-1,10-phenanthroline (*p*-MeO-Phen), silver could function as an n-dopant stronger than cesium and could be used to fabricate organic light-emitting diodes with higher performance than the cesium-doped control device.

[1] Key Lab of Organic Optoelectronics, Department of Chemistry, Tsinghua University, 100084 Beijing, China. [2] College of Chemistry, Sichuan University, 610064 Chengdu, Sichuan, China. [3] Center for Flexible Electronics Technology, Tsinghua University, 100084 Beijing, China. Correspondence and requests for materials should be addressed to L.D. (email: duanl@mail.tsinghua.edu.cn)

Organic semiconductors have shown great potential in a wide variety of optoelectronic devices, such as organic light-emitting diodes (OLEDs), organic solar cells (OSCs), and organic thin-film transistors (OTFTs)[1–4]. However, the transport properties of pristine organic semiconductors are extremely poor compared with their inorganic counterparts due to the low intrinsic carrier concentration and the weak molecular interaction. Controlled and stable doping is essential to increase film conductivity and create ohmic contacts at cathode interfaces. Though it is challenging to find suitable n-dopants for organic semiconductors with small electronic affinities (EAs)[5,6], such as electron transport materials (ETMs) in OLEDs, which generally require dopants with low ionization energies (IE), a range that was thus far the domain of very reactive alkali metals such as Li[7], Cs[8], and a few easily oxidized molecular compounds such as $W_2(hpp)_4$[9–11]. To avoid the use of these air-sensitive materials, a subtractive method has been proposed to release n-dopants in situ by using their air-stable precursors, such as widely used inorganic precursors (Li$_3$N[12], Cs$_2$CO$_3$[13], CsN$_3$[14]) and a few organic precursors (o-MeO-DMBI-I[15–18]). However, the precursor approach is always accompanied by undesired side products and massive out-gassing[19]. Recently, cleavable dimers of highly reducing organometallic species were introduced as a class of byproduct-free n-dopants for organic semiconductors, though harsh treatments such as UV-irradiation are required to overcome the thermodynamic limit and activate the dimer for efficient n-doping of ETMs in OLEDs[20]. In industry, dispensers of reactive metals are widely used to guarantee a safe and controllable deposition of n-dopants, albeit the reactive metals deposited onto the walls of the vacuum chamber are still hazardous[21]. Moreover, metal-migration of the reactive metals may severely harm the device efficiency and long-term stability[22]. Therefore, a stable and simplified way to overcome these limitations is highly desired to promote the further application of n-dopants in optoelectronic devices[9,19,23].

It has been reported that the interaction between silver and electron transport layers, such as bathocuproine (BCP) or 1,10-phenanthroline (BPhen), could improve electron injection[24,25]. Yoshida[26] proposed that the decreased lowest unoccupied molecular orbital (LUMO) level of the formed Ag-BCP complex is the main reason for the improved electron injection. Yang and Sun[27] suggested that the decreased electron injection barrier is due to the localized surface plasmonic resonance of metal nanoparticles. Since the mechanism has not been clarified yet, further investigations are needed.

Here, we firstly studied the interaction between Ag and BPhen molecules, and uncovered that BPhen tends to interact with Ag cations (Ag$^+$) rather than Ag atoms in the film, forming strong coordination bonds and stable complexes of [Ag(BPhen)]$^+$ and [Ag(BPhen)$_2$]$^+$. Based on the above understanding, we propose a byproduct-free additive method to produce n-dopants. By doping air-stable metals in organic ligands with chelating cites, the strong coordination of organic ligands with metal ions would push the equilibrium between metals and metal ions to the forward direction with free electrons more-readily to be released. In this way, the ionization energy of air-stable metals could be remarkably reduced and with proper design of the ligand structures, air-stable metals could function more efficiently as n-dopants than alkalis and improve the performance of OLEDs.

## Results

### N-doping mechanism between metals and electron chelating ligands

The mechanism is shown in Fig. 1a. As shown in Fig. 1b, alkalis have rather low ionization energies of below 5.5 eV, thus having strong donating ability and being widely used as n-dopants in organic electronics, while Ag has a high ionization energy of 7.6 eV thus it is air-stable and traditionally not being considered as an n-dopant[7,13,14,28,29]. As calculated (Fig. 1a), the ionization energy of Ag is 8.0 eV (Ag $\rightarrow$ Ag$^+$ + e$^-$, 8.0 eV), but when Ag is doped into BPhen, its ionization energy would decrease rapidly to 4.4 eV (Ag + BPhen $\rightarrow$ [Ag(BPhen)]$^+$ + e$^-$, 4.4 eV) or even 2.7 eV (Ag + 2BPhen $\rightarrow$ [Ag(BPhen)$_2$]$^+$ + e$^-$, 2.7 eV) which is lower than that of cesium, due to the irreversible coordination between Ag$^+$ and BPhen and the formation of stable complexes (Ag$^+$ + BPhen $\rightarrow$ [Ag(BPhen)]$^+$, −3.6 eV and [Ag(BPhen)]$^+$ + BPhen $\rightarrow$ [Ag(BPhen)$_2$]$^+$, −1.7 eV).

To reveal the n-doping process between Ag and BPhen, the internal chemical state of Ag in BPhen is firstly studied using X-ray photoelectron spectra (XPS) shown in Fig. 2a and auger electron spectra (AES) (Supplementary Fig. 1). The kinetic energy of auger electron for Ag element in the doped film is about 349.0 and 354.3 eV, while that of pristine Ag atom is only about 351.9 and 357.8 eV. And the core level of Ag 3d shifts towards higher binding energy from 374.3 and 368.3 eV to 374.8 and 368.8 eV. Thus, compared with standard Ag samples (Supplementary Fig. 1), we find that Ag atom is reduced to its oxidized form (Ag$^+$) when doped into BPhen film, indicating the efficient n-doping process between Ag and BPhen. And as shown from mass spectrum (Fig. 2b), except for monomer and oligomer of BPhen molecules, the doped film has another two strong signals at 438.9 and 770.6, indicating the formation of [Ag(BPhen)]$^+$ (m/z = 440.3) and [Ag(BPhen)$_2$]$^+$ (m/z = 772.9) in the doped film. As calculated, when Ag is freshly deposited into BPhen and the interaction between Ag and BPhen is only 12.3 kJ/mol, which belongs to Vander Waals' force and is a rather weak interaction. But if BPhen gets close to Ag$^+$, it would chelate strongly with Ag$^+$ to form a strong coordination bond, with over tenfold high bond energy of 173.1 kJ/mol. Although the tendency to form Ag$^+$ is negligible for pristine Ag, in the presence of BPhen, the irreversible coordination interaction between Ag$^+$ and BPhen would push the equilibrium between Ag and Ag$^+$ to the forward direction with more Ag$^+$ formed. As shown in the XPS of pristine BPhen and Ag-doped BPhen films, the core level of C 1s for BPhen shifts toward higher

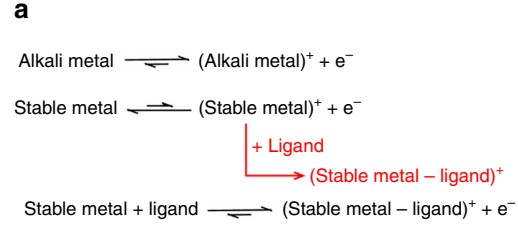

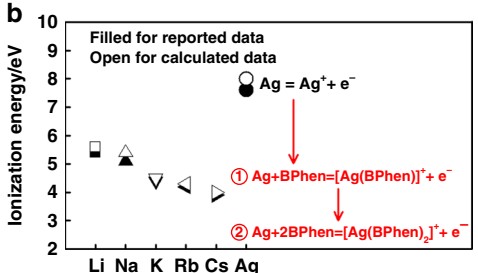

**Fig. 1** An additive method to produce n-dopants. **a** The n-doping mechanism between metals and electron-chelating ligands. **b** The experimental and calculated ionization energies of different metals: filled for reported data and open for calculated data

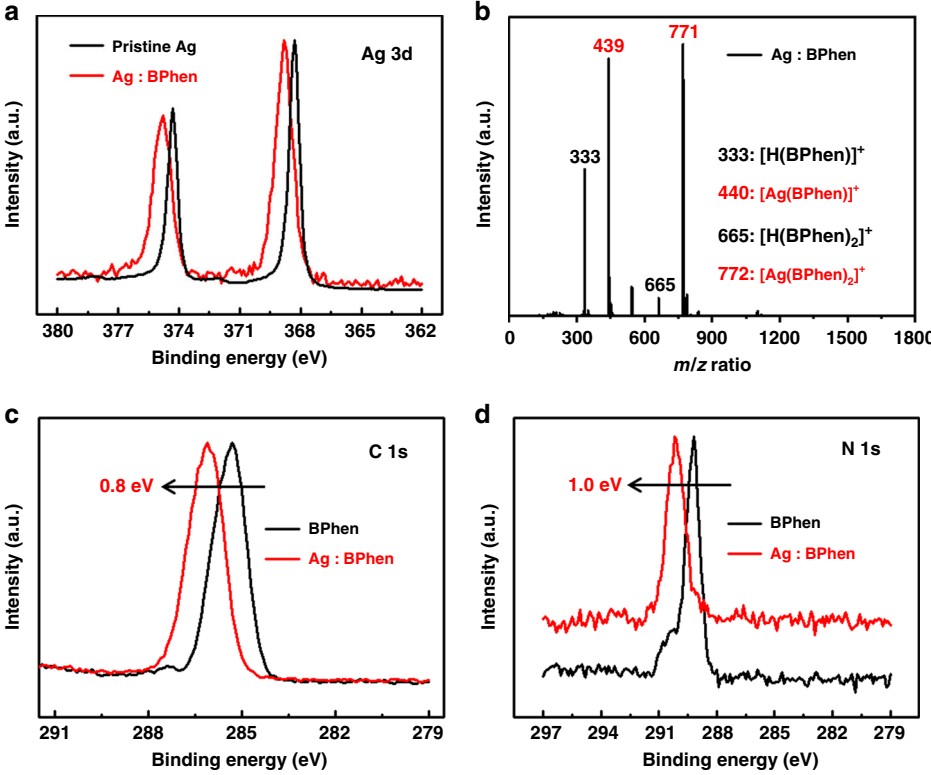

**Fig. 2** Characterization of the interaction between Ag$^+$ and BPhen. **a** XPS of Ag $3d$ core levels for pristine Ag and Ag-doped BPhen films. **b** Time-of-flight mass spectrum of Ag-doped BPhen film. **c**, **d** XPS of C $1s$ and N $1s$ core levels for pristine BPhen and Ag-doped BPhen films

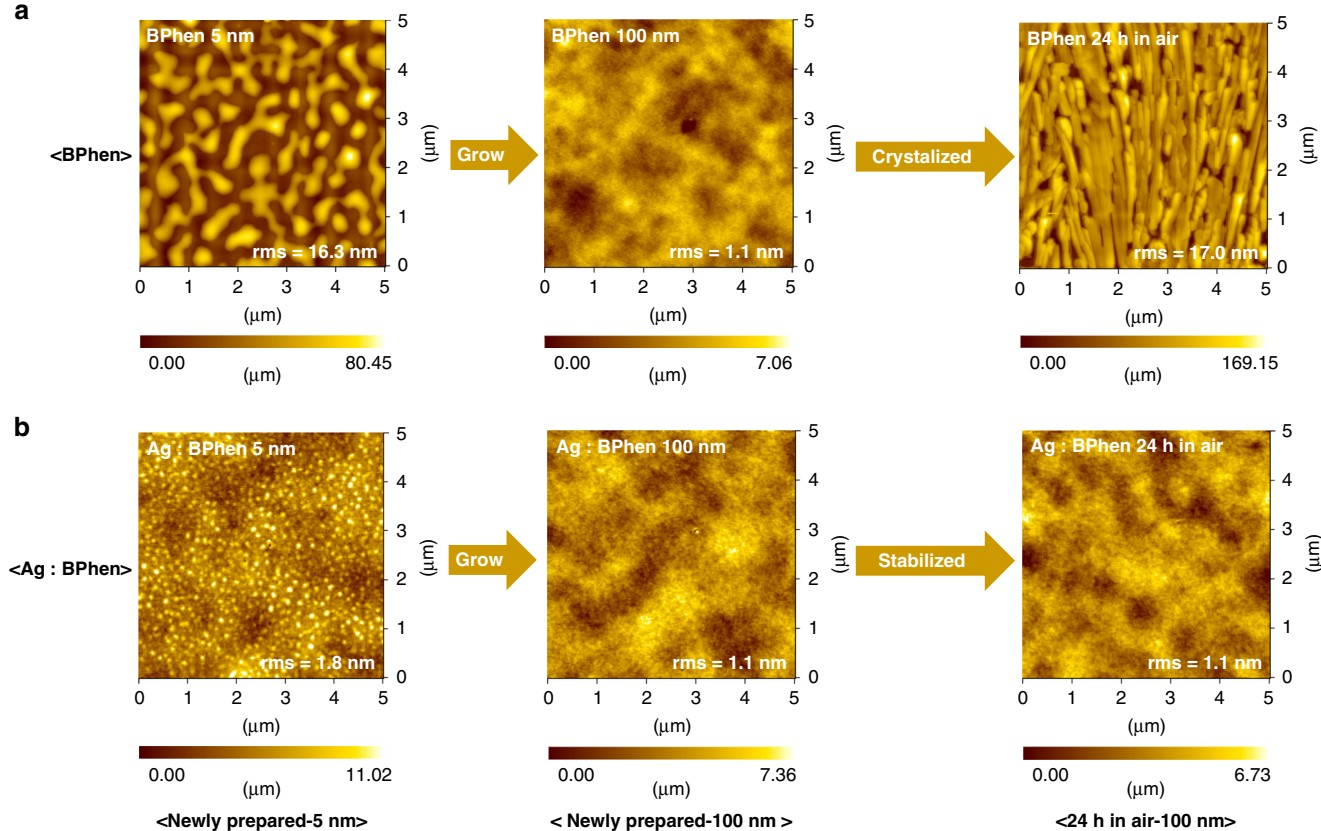

**Fig. 3** Comparison of stability between pristine BPhen and Ag-doped BPhen. The morphology of **a** pristine BPhen and **b** Ag-doped BPhen films measured right after and 24 h after the films preparation

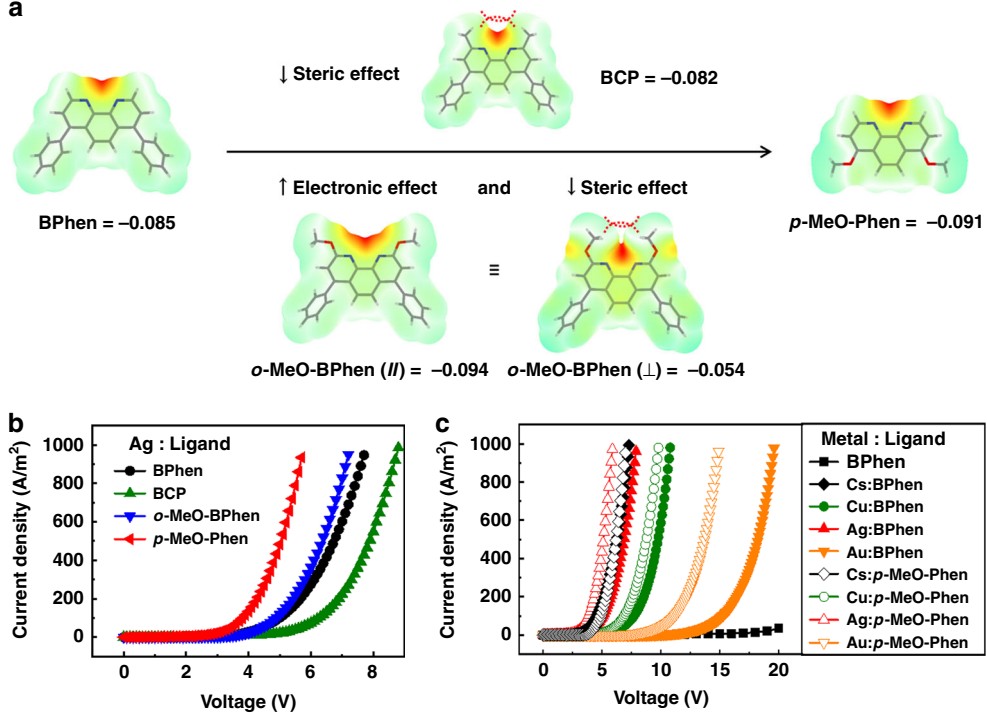

**Fig. 4** Ligands design based on theoretical calculations and device performances. **a** Electrostatic potential surfaces of different organic ligands and the maximum electrostatic potential measured around the molecules. **b**, **c** Current density–voltage characteristics of EODs using different organic ligands doped with air-stable metals, such as Cu and Ag Au. And Cs is also used for comparison. Device structure: ITO/BPhen (100 nm)/pristine or n-doped organic layer (5 nm)/Al. The doping ratio is 20 % by volume. $Cs_2CO_3$ is used as the precursor for Cs[29,36]

binding energy by 0.8 eV (Fig. 2c) and the core level of N 1s shifts toward higher binding energy by even 1.0 eV (Fig. 2d). Thus, in the Ag-doped BPhen film, the upward shift of binding energy for N 1s is 0.2 eV is higher than that of C 1s due to the most nucleophilic quality around the molecule of N atoms, which are considered as the main binding sites with $Ag^+$. The existence of coordination and doping effects could also be demonstrated from the nuclear magnetic resonance (NMR) spectra (Supplementary Fig. 2). Compared with pristine BPhen, the coordination interaction between BPhen and $Ag^+$ would largely reduce the electron density of 1,10-phenanthroline ring, thus leading to the upward chemical shifts of the hydrogen atoms in 1,10-phenanthroline ring. For example, the chemical shifts are all moved upwards from 9.25, 7.86, and 7.60 ppm to 9.28, 7.90 and 7.72 ppm for H2, H5, and H3 of BPhen, respectively. However, for Ag-doped BPhen, the upward movements of H3 and H5 are only about 0.2–0.3 ppm, which are smaller than $Ag^+$-doped one. And as for H2 which locates closer to nitrogen atom, the chemical shift moves in the opposite direction from 9.25 ppm to 9.24 ppm, which means that although there exists a coordination interaction between BPhen and $Ag^+$, the strong n-doping effect between Ag and BPhen could greatly increase the electron density in 1,10-phenanthroline ring. Thus, it leads to the downward chemical movement of H2 and only a slightly upward chemical movement of H3 and H5.

**Stability of n-doped films**. Intriguingly, the strong coordination bond between $Ag^+$ and BPhen leads to largely improved film stability. As shown in Fig. 3, when a thin layer of BPhen (5 nm) is deposited on a smooth ITO substrate, it tends to form agglomerates with island-like geometry due to its flat molecular structure. As more BPhen is deposited, the agglomerates tend to grow higher and bigger, then form a continuous film. The molecular agglomeration would lead to a strong crystallization after the film

is prepared, as can be seen from the morphology change when pristine BPhen film is placed in air for 24 h, which shows a largely increased film root-mean-square (rms) from 1.1 to 17.0 nm (Fig. 3a). The film crystallization would cause a severe interface separation and has been recognized harmful for devices. However, when Ag is deposited into BPhen, any $Ag^+$ formed via a weak dynamic equilibrium would react with BPhen to form metal–organic complexes, which is good for film growth and largely restrains the molecular agglomeration (Fig. 3b). Thus, with an eliminated molecular agglomeration by interaction with $Ag^+$, the Ag-doped BPhen film is more stable and little changes are observed as time passes. It has been reported that pristine BPhen is not an ideal electron transport material for a stable device. Here, with a strong interaction between $Ag^+$ and BPhen, the film stability could be largely improved.

**Ligand design and n-doping efficiency**. Then electron-only devices (EODs) are used to compare the electron injection properties of non-doped and Ag-doped BPhen (Fig. 4 and Supplementary Fig. 3). Without any n-dopant, electrons are difficult to be injected from the cathode because of large electron injection barrier, leading to the low current density for EOD. Then depositing Ag into BPhen with an optimized doping concentration of 20% at cathode interface could immediately boost the device current density from 0.02 to 100 $A/m^2$ at a 5 V bias, comparable to that of the device doped with Cs and almost four orders of magnitude higher than that of the undoped control device. The Ag-doped BPhen film is an efficient electron injection layer that even under a high-work-function cathode such as Ag, a similar high current density can still be maintained (Supplementary Fig. 3). But when Ag is moved from the bulk of BPhen film to its surface, the current density of EOD is largely decreased (Supplementary Fig. 3).

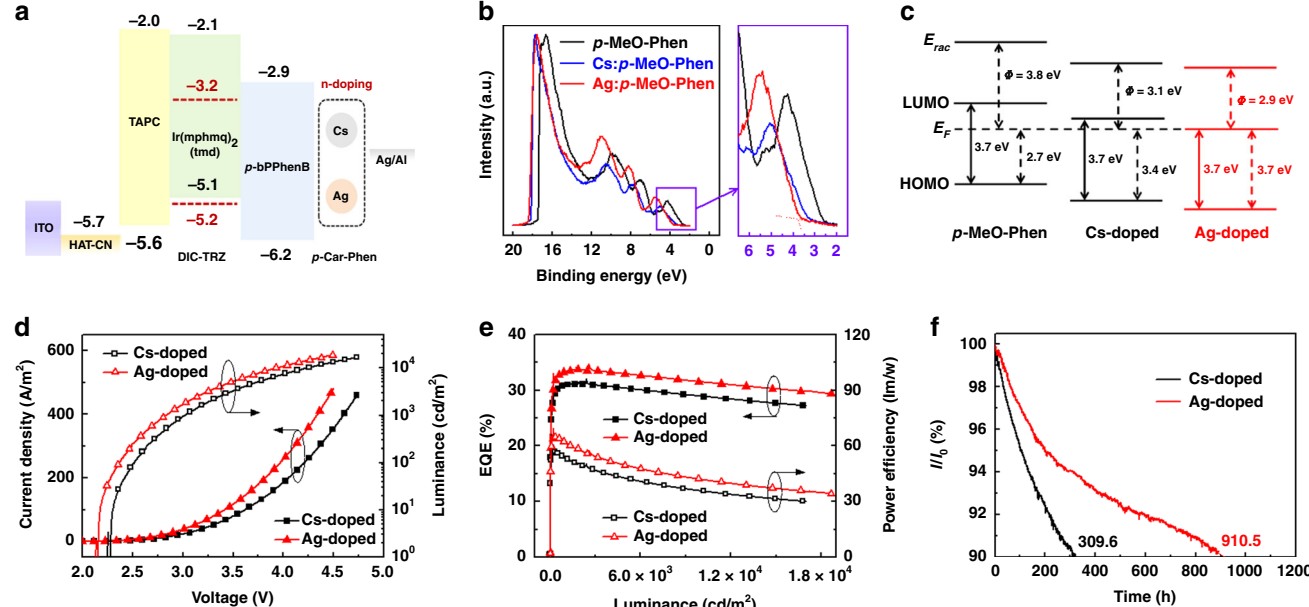

**Fig. 5** Energy diagram and performance of OLEDs with Ag or Cs as the n-dopant. **a** The device structures of OLED devices. **b** The UPS spectra and **c** the schematic energy-level diagrams of pristine, Cs-doped, and Ag-doped p-MeO-Phen films for comparison. **d** The current density-luminance-voltage, **e** the current efficiency-power efficiency-luminance, and **f** the luminance-deterioration of OLED devices using Cs and Ag as n-dopants for comparison. The structures of device are ITO/HAT-CN (5 nm)/TAPC (120 nm)/Ir(mphmq)$_2$(tmd):DIC-TRZ = 3.5% (35 nm)/p-bPPhenB (40 nm)/Cs or Ag-doped p-MeO-Phen (5 nm)/Ag (10 nm)/Al (100 nm). HAT-CN is 1,4,5,8,9,11-hexaazatriphenylene hexacarbonitrile, NPB is N,N′-di(naphthalene-1-yl)-N,N′-diphenyl-benzidine, TAPC is 4,4′-cyclohexylidenebis[N,N-bis(4-methylphenyl)aniline], Ir(mphmq)$_2$(tmd) is (bis(4-methyl-2-(3,5-dimethylphenyl) quinoline)) Ir(III) (tetramethylheptadionate), DIC-TRZ is 2,4-diphenyl-6-bis(12-phenylindolo)([2,3-a]carbazole-11-yl)-1,3,5-triazine, and p-bPPhenB is 1,4-bis(2-phenyl-1,10-phenanthrolin-4-yl)benzene

To study the effect of the chelating property of the ligand on the n-doping efficiency, different organic ETMs are compared in EODs (Fig. 4 and Supplementary Fig. 3). When Ag is doped into 1,3,5-tris(1-phenyl-1H-benzimidazol-2-yl)benzene (TPBi) with a similar energy level as BPhen but negligible chelating property, no improvement in electron injection can be observed and the current density of EOD keeps at a rather low level. So the coordination reaction between dopant and matrix material is crucial for the n-doping process. The molecular electrostatic potentials are calculated and used to detect the nuclear and electronic charge distribution among the ligands. As shown in Fig. 4a, nitrogen atoms around the molecules exhibit the most nucleophilic quality, and are considered as the main binding sites with Ag$^+$. Thus, we could introduce electron-donating groups like methoxyl- at the ortho- or para- position of nitrogen atoms to increase their nucleophilic quality and improve the doping efficiency. For example, o-MeO-BPhen and p-MeO-Phen, with stronger binding sites than BPhen, could both achieve higher electron injection efficiencies. However, except for electronic effect, steric effect is also a crucial factor for metal binding[30,31]. Thus, introducing a methoxyl-group at the ortho- position of nitrogen atom is not the best way to improve n-doping efficiency for that the methoxyl-group is likely to rotate around the nitrogen atom and would introduce a steric hindrance for metal binding. Among the all designed ligands, p-MeO-Phen has a high electrostatic potential around nitrogen atoms together with negligible steric hindrance for metal binding. Hence, it exhibits improved current density in EODs from ~100 to 483 m/A$^2$ at a 5 V bias compared with device using BPhen as the ligand (Fig. 4b), indicating a much higher n-doping efficiency. However, when Cs is doped into p-MeO-Phen, no obvious improvement in current density for EOD could be achieved compared to that of BPhen-containing device due to the limited coordination tendency of Cs$^+$ with the organic host (Fig. 4c). Besides BPhen derivatives and Ag,

other ETMs with chelating sites such as 2,7-di([2,2′:6′,2″-terpyridin]-4′-yl)-9,9′-spirobifluorene (27-TPSF, Supplementary Fig. 3) and air-stable metals such as Cu and Au (Fig. 4c) also show great ability to form efficient n-doped layers and enhance electron injection. It is believed that with proper design of ligand structures, the n-doping efficiency could be further improved.

**Fabrication and performance of OLEDs.** Here, we further compared Ag and Cs-doped p-MeO-Phen as electron injection layers in OLEDs with high-work-function cathode of Ag (Fig. 5a). Without any n-dopant, electrons are hardly injected from cathode, thus leading to rather high driving voltage and low power efficiency (Supplementary Fig. 4). The ultroviolet photoelectron spectroscopy (UPS) shown in Fig. 5b and Supplementary Fig. 5 indicates that the work function of Ag cathode could be largely reduced from 4.2 to 2.8 eV when Ag-doped p-MeO-Phen is deposited, which is 0.2 eV lower than reactive Cs-based film due to a stronger n-doping effect. With highest occupied molecular orbital (HOMO, −6.3 eV) measured from UPS spectra and LUMO (−2.6 eV) of p-MeO-Phen measured using the low-energy inverse photoemission spectroscopy (LEIPS), the energy diagram of could be drawn as Fig. 5c[32]. The edge of HOMO for pristine p-MeO-Phen film is located 2.7 eV below the Fermi level, and it is increased by about 0.7 eV when Cs is doped. As for the Ag-doped p-MeO-Phen film, the Fermi level raises up by another 0.3 eV towards its LUMO, demonstrating a further decreased electron injection barrier and an Ohmic contact. As a result, the turn-on voltage of OLED with Ag-doped p-MeO-Phen is decreased from 2.3 to 2.1 V, compared with the control device using Cs as an n-dopant (Fig. 5d). And the voltages required to get 1000 and 10,000 cd/m$^2$ are only 2.8 and 3.9 V for the Ag-doped device, which are the lowest for red OLEDs ever reported and valuable for real applications. Due to small efficiency roll-off by improved charge balance, the external quantum efficiency

(EQE) and power efficiency of the Ag-doped device are 33.4% and 61.7 lm/W at 1000 cd/m$^2$ and still maintain 31.5% and 41.9 lm/W at 10,000 cd/m$^2$, respectively, all higher those of the Cs-doped device (30.9% and 53.9 lm/W at 1000 cd/m$^2$ and 28.9% and 35.7 lm/W at 10,000 cd/m$^2$) and representing the best overall performance of red OLEDs in literatures (Fig. 5e). Moreover, the operation stability is also largely enhanced. $T_{90}$, defined as the time when the actual luminance decays to 90% of the initial luminance, is prolonged from 309.6 to 910.5 h at 1000 cd/m$^2$ (Fig. 5f). And both of the devices give typical red emission at 608.0 nm from Ir(mphmq)$_2$(tmd), indicating that $n$-dopants added in the devices have negligible influence for light emission (Fig. 5f). The histograms of EQEs$_{max}$ and lifetimes measured from different devices are shown in Supplementary Fig. 6. It is evident that $n$-doped film based on air-stable metals may pave the way for the easy fabrication of highly efficient and stable OLEDs, which is crucial for the commercial displays.

## Discussion

In this work, we provide an additive method to activate air-stable metals such as silver to be efficient $n$-dopants, whose reduction potentials are even beyond the reach of cesium, via in situ reaction with well-designed multi-functional ETMs. It is further demonstrated that an Ag-doped $p$-MeO-Phen layer could produce low-work-function electrodes with a variety of materials, from metals to metal oxides to organics (Supplementary Fig. 7), similar to PEI and PEIE[33–35]. The above results suggested that our method is not only meaningful for organic semiconductors, but may have great potential for other semiconductors, such as perovskites, carbon nanotubes, graphenes and inorganic two-dimensional materials.

## Methods

**Theoretical calculations**. The geometrical and electronic property were calculated with Gaussian 03 program package. The calculations were optimized by means of B3LYP (Becke Three Parameters Hybrid Functional with Lee-Yang-Perdew correlation functions). The ionization energies were calculated using the B3LYP/6–311++G (2df,2p) electronic energy and the zero-point vibrational energy and thermal corrections (0 to 298 K) obtained at the B3LYP/6–31+G(d) level. The electrostatic potential surfaces were visualized and drawn using Gaussview software.

**Film and device characterization**. The films were prepared by conventional thermal deposition inside a high vacuum chamber below $10^{-4}$ Pa on ITO substrates with sheet resistances about 20 Ω/square. Metals and ETMs were co-evaporated inside a high vacuum chamber at $2 \times 10^{-4}$ Pa. The coating rate of ETM was 1 Å/s, and for metal such as Ag, the coating rate varied from 0.05 to 0.3 Å/s according to the doping concentration. Absorption spectra were recorded with an UV-vis spectrophotometer (Jobin Yvon, FluoroMax-3). X-ray photoelectron spectroscopy (XPS) were measured on ULVAC-PHI, PHI Quantera SXM. The morphologies were studied by atomic force microscope (AFM) using Seiko instrument SPA 400 AFM system.

## Data availability

The data that support the plots within the paper are available from the corresponding author upon reasonable request.

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

## Acknowledgements
We would like to thank the National Key Basic Research and Development Program of China (Grant Nos. 2016YFB0400702, 2016YFB0401003, 2017YFA0204501), the National Basic Research Program of China (Grant No. 2015CB655002), and the National Natural Science Foundation of China (Grant Nos. 51525304, 61474069, and U1601651) for financial support. We are grateful to Prof. C. W. Tang for his valuable comments and suggestions. We would like to thank Profs. Chihaya Adachi and Toshinori Matsushima of Kyushu University for the measurement of the LUMO level of *p*-MeO-Phen using low-energy inverse photoemission spectroscopy (LEIPS) and valuable discussions. We also would like to thank Prof. Jang Hyuk Kwon and Prof. Lixin Xiao for providing the electron transport material of *p*-bPPhenB and 27-TPSF.

## Author contributions
Z.B. designed and carried out most part of the experiments. G.D. studied the electrical properties of the doped films. Z.B. and D.Z. designed the OLED devices. Z.L. and P.W. measured the EODs. R.S. carried out the NMR measurements. Z.B., Y.Q. and L.D. analyzed data and wrote the manuscript. L.D. conceived and supervised the project.

## Additional information

**Competing interests:** The authors declare no competing interests.

