## [Peer Review File · Nature Communications]

Reviewers' comments:

Reviewer #1 (Remarks to the Author):

The referee has now seen this manuscript three times including this version. The first two were for a submission to Science. In the process of revising their manuscript following the two Science rounds of review, the authors have somewhat improved their paper. Yet, although the topic and the potential applications are important, the paper is still not ready for publication.

The first two reviews for Science are included below to refresh the authors' memory. Before the paper can be recommended for acceptance, the points listed after these two reviews must be addressed.

First Science review:

The manuscript addresses an interesting issue for organic electronics, i.e. the formation of strong n-dopants from inert metal atoms, and also potentially resolves a long-standing controversy on the mechanism that makes BPhen a good electron injection layer when contacted with Ag. These two points are related here to an irreversible coordination reaction between Ag atoms and BPhen molecules, leading to the formation of a stable complex like [Ag(BPhen)] with low ionization energy, and which acts as an efficient electron donor. The authors present a range of characterizations of the system.

The first problem with the manuscript is the claim made in the middle of p. 2 of a "universal byproduct-free additive method". While the byproduct-free part of the claim is true, the claim of universality seems to be somewhat of a stretch. The authors demonstrate that Ag, Cu, and to some extent Au, lead to similar results, at least with p-MeO-Phen, however, the methods appears to be limited in this paper to a specific class of organic molecules. To contrast to another "universal" method to produce low work function electrodes (see Zhou et al. Science, 336, 327 (2012)), the latter was applicable to all materials investigated, from metals to metal oxides to organics. The claim of universality made here is therefore unwarranted.

In Fig. 2, the authors show an Auger electron spectrum of Ag in the complex. Why use AES here, whereas XPS, which more directly give chemical shifts, is used for C and N? where is the comparison with neutral Ag? Fig. 2C and 2D show core level shifts to higher binding energy, that could be attributed to the loss of electron by the Ag-BPhen complex (chemical shift), or to the shift of the Fermi level in the BPhen gap (electrostatic shift). Which is it? If it is the former, as claimed by the authors at the bottom of p. 3, and given that only few Ag atoms have been added to the BPhen layer, the C and N core levels should show broadening or multiple components.

The TOF measurements of electron mobility must be shown and the mobility-determination must be detailed. This is a very important issue. The statement made at the top of p. 5 just before Fig. 3, which attribute the increase in mobility to a reduced hopping distance is unsubstantiated.

Increased mobility can be due to a number of things, e.g. trap filling.

The claim of efficient doping made throughout the manuscript should also be better quantified.

Yes, the current density increases by several orders of magnitude, but this has been seen in multiple experiments of organic semiconductor doping. C-V measurements on BPhen or p-MeO-Phen layers with controlled amounts of Ag would go a long way toward resolving this issue.

The energy diagrams presented in Fig. 5 show that the Fermi level in the Phen derivative approaches very close to the LUMO. Actually, the figure claims that the Fermi level is at the LUMO in the Ag-doped layer. This based on a gap of 3.4 eV for this material. How is this gap known? As shown in so many previous publication, the gap used here cannot be the optical gap, in view of the sizable exciton binding energy in these molecular materials. Looking at previous publications in which the single particle gap was determined by direct and inverse photoemission spectroscopy, this gap is 4.4 eV for BPhen. It is high unlikely that this gap would be 3.4 eV for the material used in Fig. 5, its LUMO being probably significantly higher. The point here is that the very favorable picture painted in the energy diagram of Fig. 5 is, at this point, unsubstantiated.

Typo:

Bottom of p. 2: the reference to Fig. 2D is wrong

For the reasons stated above, the referee does not think that this paper can be accepted in

Science.

Second Science review:

The referee acknowledges the effort by the authors, however some of the responses are not satisfactory and the manuscript cannot be accepted as is.

First of all, the referee would appreciate a detailed and systematic list of everything that has been changed in the manuscript, with new parts of the manuscript clearly outlined or highlighted.

Lacking this makes it far more difficult to see how effective the authors' response is.

Regarding the shift of the N and C core levels, the authors' answer is not satisfactory, as they do not address the Fermi level shift. Obviously, the Fermi level shifts upwards in the gap of the p-MeO-Phen, and this must have a key role in the shift of the core levels toward higher binding energy. The authors must determine which parts are due to chemical shift and to Fermi level shift. The authors also do not address the question of the referee regarding the broadening of the C and N core levels.

The response about the increase in mobility is also not satisfactory. How is the activation energy calculated? The original manuscript mentioned "a reduced hopping distance" as the cause for increased mobility, and now the response mentions a "reduced hopping energy" as well as trap filling. Which is it? It seems that the authors are giving answers without justification either.

Publication in Science requires that all points made in a paper be very clear and fully substantiated. At this point, this manuscript has not reached this level.

As for the Nature Materials submission:

The question regarding the use of Auger electron spectroscopy, rather than XPS, and the issue regarding the N and C core levels raised in the first and second Science reviews remains to be addressed in the present manuscript. The authors must explain these points.

The claim made in the original paper of "universal byproduct-free additive method" (see first Science review) is now back in the present paper. This is not acceptable. See my comments in the first review above.

Finally, the English remains poor, and must be corrected, preferably by a native speaker.

Reviewer #2 (Remarks to the Author):

The discovery of an air-stable n-dopant for organic semiconducting layer is crucial toward to future development of organic electronics. This manuscript presents very interesting data on air-stable metal stronger N-dopant than Alkalis. Interestingly, the authors use silver as N-type dopant for phenanthroline derivatives. The present data in this manuscript indicate that the combination can lead to an enhancement of electrical conductivity in the co-deposited film due to an electron release after the coordination reaction between silver and phenanthroline derivatives. Despite the fact that previous studies have already focused on the interaction of alkali metal such as cesium with organic molecules, an approach of the utilization of silver appears to be missing. Overall, this manuscript has a possibility for the publication in Nature communications. However, in my opinion, a major revision of manuscript is needed before it can be accepted for publication.

Some more major considerations:

1) One serious problem in this manuscript is that the details of the coordination reaction mechanism are not clear. What is a driving force to proceed a coordination reaction between silver and phenanthroline derivatives? In other word, how to create Ag⁺ at initially? More deep explanation on the mechanism of the reaction should be given with theoretical calculation results. In addition, the authors also should have mentioned the reaction efficiency.

2) Is it realized Ohmic contact between co-deposited layer (Ag:organic) / electrode interface?

3) If the author's mentioned mechanism is true, an electron injection barrier at an interface

between pristine BPhen / pristine silver should be also negligible small according to an interfacial carrier doping. The electrical characteristics (J-V and UPS data) at the interface should be given.

4) The device operational lifetime data of the OLED based on Cs-doped ETL should be given to evaluate the OLED performance of the OLED with Ag-doped ETL.

Technical comments:

- 1) The EL spectrum of each OLEDs should be given in Supporting Information to confirm a cavity effect.
- 2) Page 6: The unit is incorrect, i.e., $\text{m/A}^2 \rightarrow \text{A/m}^2$

Detailed Response to Reviewer

Reviewer 1

The referee has now seen this manuscript three times including this version. The first two were for a submission to Science. In the process of revising their manuscript following the two Science rounds of review, the authors have somewhat improved their paper. Yet, although the topic and the potential applications are important, the paper is still not ready for publication.

The first two reviews for Science are included below to refresh the authors' memory. Before the paper can be recommended for acceptance, the points listed after these two reviews must be addressed.

(1) The question regarding the use of Auger electron spectroscopy, rather than XPS

Author reply:

→Thank you for your good questions and nice comments. According to your suggestion, we have further measured the X-ray photoelectron spectra (XPS) to study the internal chemical state of Ag and reveal the n-doping process between Ag and BPhen as shown in Fig. 2A.

Figure 2. (A) XPS and (B) AES of Ag 3d core levels for pristine Ag and Ag-doped BPhen films. (C) The binding energy of Ag 3d and the kinetic energy of Ag MNN for different Ag samples.

The kinetic energy of auger electron for Ag element in the doped film is about 349.0 eV and 354.3 eV, while that of pristine Ag atom is only about 351.9 eV and 357.8 eV. And the core level of Ag 3d shifts towards higher binding energy from 374.3 eV and 368.3 eV to 374.8 eV and 368.8 eV. Thus compared with standard Ag samples, we find that Ag atom is reduced to its oxidized form (Ag⁺) when doped into BPhen film, indicating the efficient n-doping process between Ag and BPhen.

(2) Regarding the shift of the N and C core levels, the authors' answer is not satisfactory, as they do not address the Fermi level shift. Obviously, the Fermi level shifts upwards in the gap of the *p*-MeO-Phen, and this must have a key role in the shift of the core levels toward higher binding energy. The authors must determine which parts are due to chemical shift and to Fermi level shift.

Author reply:

→ Thank you for your nice comments. As shown from XPS and AES spectra, we demonstrate that Ag atom is reduced to its oxidized form (Ag^+) when doped into BPhen film, indicating the efficient n-doping process between Ag and BPhen. Just as you mentioned, this n-doping process would lead to the upward shift of Fermi level in the gap, and has a key role in the shift of the core levels toward higher binding energy. Thus as shown in the XPS spectra, the core level of C 1s and N1s of BPhen shifts towards higher binding energy. On the other hand, as demonstrated from mass spectrum and theoretical calculations, the coordination complexes of $[\text{Ag}(\text{BPhen})]^+$ and $[\text{Ag}(\text{BPhen})_2]^+$ could be formed in the Ag-doped BPhen film, which is also favorable for the upward shift of C1s and N1s. We admit that it is hard to determine which parts are due to chemical shift and to Fermi level shift. Here, we find that the upward shift of binding energy for N 1s is 0.2 eV is higher than that of C 1s. This is due to the most nucleophilic quality around the molecule of N atoms which are considered as the main binding sites with Ag^+ and may be the clue for chemical shift *via* coordination reaction.

Figure 2. (C)-(D) XPS of C 1s and N 1s core levels for pristine BPhen and Ag-doped BPhen films.

(3) The response about the increase in mobility is also not satisfactory. How is the activation energy calculated? The original manuscript mentioned “a reduced hopping distance” as the cause for increased mobility, and now the response mentions a “reduced hopping energy” as well as trap filling. Which is it?

Author reply:

→Thank you for your good question. With careful consideration, we make some adjustment for this paper to make it more clear. In this paper, we aim to propose a byproduct-free additive method to process n-dopants and use the doped layer to improve the OLED performance by reducing electron-injection barrier and achieve Ohmic contact. Thus in the revised version, we drop the transporting part and concentrate more on analysis of electron injection upon doping.

Reviewer 2

The discovery of an air-stable n-dopant for organic semiconducting layer is crucial toward to future development of organic electronics. This manuscript presents very interesting data on air-stable metal stronger N-dopant than Alkalis. Interestingly, the authors use silver as N-type dopant for phenanthroline derivatives. The present data in this manuscript indicate that the combination can lead to an enhancement of electrical conductivity in the co-deposited film due to an electron release after the coordination reaction between silver and phenanthroline derivatives. Despite the fact that previous studies have already focused on the interaction of alkali metal such as cesium with organic molecules, an approach of the utilization of silver appears to be missing. Overall, this manuscript has a possibility for the publication in Nature communications. However, in my opinion, a major revision of manuscript is needed before it can be accepted for publication.

(1) One serious problem in this manuscript is that the details of the coordination reaction mechanism are not clear. What is a driving force to proceed a coordination reaction between silver and phenanthroline derivatives? In other word, how to create Ag⁺ at initially? More deep explanation on the mechanism of the reaction should be given with theoretical calculation results. In addition, the authors also should have mentioned the reaction efficiency.

→Thank you for your nice question. In this manuscript, we aim to propose a byproduct-free additive method to produce n-dopants via coordination reaction. This idea is mainly inspired from the traditional method of dissolving gold with aqua regia, in which nitric acid is used as a strong oxidant while hydrochloric acid is such an important chelating reagent to reduce the reduction potential of gold by forming [AuCl₄]⁻ (reaction 1: Au+HNO₃+4HCl=H[AuCl₄]+NO+2H₂O). Here we show that n-dopants can be formed via an ideal additive method by *in-situ* irreversible coordination reaction between inert-metal ions and electron-transporting materials, which could push the equilibrium reaction between metal and metal ion to the forward direction with free electrons released and achieve efficient n-doping process. Just as the reaction 1 shown above, when Ag is deposited into BPhen, any Ag⁺ formed via a weak dynamic equilibrium would react with BPhen to form metal-organic complexes. Although the tendency to form Ag⁺ is negligible for pristine Ag, in the presence of BPhen, the irreversible coordination interaction between Ag⁺ and BPhen would push the equilibrium between Ag and Ag⁺ to the forward direction with more Ag⁺ formed. However, as the n-doping process is realized *in situ*, it is hard to decide the reaction efficiency of the doped film in vacuum at this stage.

(2) Is it realized Ohmic contact between co-deposited layer (Ag:organic) / electrode interface?

→Thank you for your valuable suggestions. As we measured from UPS spectra and discussed in the last part of the paper: for the Ag-doped *p*-MeO-Phen film, the Fermi level raises up by another 0.2 eV towards its LUMO, demonstrating a further

decreased electron injection barrier from 1.3 to 0.3 eV and an Ohmic contact.

Fig. The schematic energy-level diagrams of pristine, Cs-doped and Ag-doped p-MeO-Phen films for comparison

(3) If the author's mentioned mechanism is true, an electron injection barrier at an interface between pristine BPhen / pristine silver should be also negligible small according to an interfacial carrier doping. The electrical characteristics (J-V and UPS data) at the interface should be given.

→Thank you for your valuable suggestions. We have further measured the electrical characteristics (J-V) for the interface doping, as shown from Fig. S2. When Ag is moved from the bulk of BPhen film to its surface, the current density of electron-only device is largely decreased but still higher than non-doped control device (Fig. S2).

Fig. Current density-voltage characteristics of electron-only devices (EODs) with different electron injection layers (W/o: ITO/ BPhen (100 nm)/ Al, interface: ITO/ BPhen (100 nm)/ Ag (10 nm)/ Al and doped: ITO/ BPhen (100 nm)/ Ag-doped BPhen = x % (5 nm)/ Al).

(4) The device operational lifetime data of the OLED based on Cs-doped ETL should be given to evaluate the OLED performance of the OLED with Ag-doped ETL.

→Thank you for your valuable suggestions. We have measured and compared the device operational lifetime of OLEDs using different n-dopants, as shown from Fig. 5E. Due to small efficiency roll-off by improved charge balance, the operation stability is also largely enhanced. T_{90} , defined as the time when the actual luminance decays to 90% of the initial luminance, is prolonged from 309.6 h to 910.5 h at 1000

cd/m².

Figure 5E. The luminance-deterioration of OLED devices using Cs and Ag as n-dopants for comparison.

(5) The EL spectrum of each OLEDs should be given in Supporting Information to confirm a cavity effect.

→Thank you for your valuable suggestions. We have measured the EL spectra of OLED devices, as shown from Figure 5F.

Figure 5F. The normalized EL curves of OLED devices using Cs and Ag as n-dopants for comparison.

(6) Page 6: The unit is incorrect, i.e., m/A² → A/m²

→Thank you for your suggestions. We have carefully revised our manuscript and correct some of the errors.

Reviewers' comments:

Reviewer #2 (Remarks to the Author):

The referee acknowledges the strong effort on the first round by the authors. Undoubtedly, the authors have presented good work. But some of the responses are not reached to the satisfied level. Considering this journal scope and this revised work, I don't recommend this paper for the publication. I recommend the authors to submit this work to the sister journal of device physics.

1) The experimental results present in this manuscript may provide some evidence for the formation of metal complex, however, the mechanism regarding the formation of the silver-Bphen complex still unclear. To reach the acceptable level for this journal, the clear explanation regarding the formation of Ag⁺ in the BPhen matrix should be given by theoretically.

Additional comments;

1) The data reproducibility, especially, EQE and device lifetime data, should provide as a histogram.

2) The OLED performance data of the device without EIL should add to evaluate the effect of the n-doping layer on the OLED performances.

Reviewer #3 (Remarks to the Author):

The paper "Making Silver a Stronger N-dopant than Cesium via in-situ Coordination Reaction for Organic Electronics" by Zhengyang Bin and colleagues reports the quite surprising observation that blending organic electron-transporting materials with silver atoms produces n-doped layers which improve injection into OLED devices significantly. The authors show by XPS/UPS and electrical measurements quite convincingly the presence of these effects and that they significantly improve devices. What is completely missing however is a proof of the generality of these effects. The specific material chosen here has in a similar form been used a long time ago for OLED but is not useful for highly stable commercial devices. From the data shown in the paper, it is not possible to conclude the generality of the effect. Although the paper deserves publication, I thus do not see it as sufficiently important to be published in Nature Comm.

Before publication, the authors should polish the language and remove the numerous typographical errors.

Reviewer #4 (Remarks to the Author):

Report NCOMMS-18-15586A-Z "Making Silver a Stronger N-dopant..." by Bin et al.

I share the opinions of the previous reviewers that the manuscript describes an interesting approach of using Ag as n-dopant for electron injection layer, and it may have the possibility of being published in Nat. Comm. The authors have made substantial improvement on the manuscript by adopting the previous reviewers' suggestions. However, more major revisions are still necessary, and the outcome of which will determine whether it is self-consistent and making the cut:

1. The energetics. In Line 90 the authors discussed the huge gain in energy of chelation of Ag. They should piece together the whole energetics, including that of Ag⁰Ag⁺+e to demonstrate the complete chemistry of the process.
2. The energy level schematics in Fig. 5B is obtained from Fig. S3B. Fig. S3B should be moved to Fig. 2 to present this important data.
3. From the right panel of Fig. S3B, it looks the HOMO position of Ag:p-MeO-Phen is 3.6 eV or

higher. They should verify it and mark on the figure how the position of the HOMO is extrapolated, and modify Fig. 5B if necessary. Furthermore, they should compare the shift of the HOMO with those of the core levels, as it is essential in doping that all energy levels of non-reactive elements shift the same amount. This analysis may also address the concern of Reviewer 1 on how to distinguish the doping induced and chemically induced energy level shift.

4. Another key to being doping instead of forming new compounds is to check if the energy gap has changed. To clarify that, the authors should add IPES data of doped samples.

5. Inconsistency between the figure and description. In Line 175, it says LEIPS, while in Fig. S3B it is IPES.

6. The process of the doping should be added to Materials and Methods.

Detailed Response to Reviewers

Reviewer 2

- (1) The experimental results present in this manuscript may provide some evidence for the formation of metal complex, however, the mechanism regarding the formation of the silver-Bphen complex still unclear. To reaches the acceptable level for this journal, the clear explanation regarding the formation of Ag^+ in the BPhen matrix should be given by theoretically.

Author reply:

→Thank you for your good questions and nice comments. To get more in-depth understanding of the doping mechanism, we provide a more direct evidence for the presence of both coordination and doping effects using nuclear magnetic resonance (NMR) spectra, as shown in Fig. S1. Compared with pristine BPhen, the coordination interaction between BPhen and Ag^+ would largely reduce the electron density of 1,10-phenanthroline ring, thus leading to the upward chemical shifts of the hydrogen atoms in 1,10-phenanthroline ring. For example, the chemical shifts are all moved upwards from 9.25, 7.86 and 7.60 ppm to 9.28, 7.90 and 7.72 ppm for H2, H5 and H3, respectively. However, for Ag-doped BPhen, the upward movements of H3 and H5 are only about 0.2-0.3 ppm, which are smaller than Ag^+ -doped one. And as for H2 which locates closer to nitrogen atom, the chemical shift moves in the opposite direction from 9.25 ppm to 9.24 ppm, which means that although there exists a coordination interaction between BPhen and Ag^+ , the strong n-doping effect between Ag and BPhen could greatly increase the electron density in 1,10-phenanthroline ring. Thus it leads to the downward chemical movement of H2 and only a slightly upward chemical movement of H3 and H5.

Fig. S2. The comparison of chemical shifts for pristine BPhen, Ag^+ -doped BPhen and Ag-doped BPhen. (For pristine BPhen (black) and Ag:BPhen (red), their films were firstly prepared by vacuum deposition, following by CDCl_3 dissolving and nuclear magnetic resonance (NMR) measurement. And for Ag^+ +BPhen (blue), AgNO_3 was added to CDCl_3 solution of BPhen for NMR measurement.)

(2) The data reproducibility, especially, EQE and device lifetime data, should provide as a histogram.

Author reply:

→ Thank you for your nice suggestion. We have added the histogram to demonstrate the device and data reproducibility as follows.

Fig. S4. The histograms of EQE_{max} and lifetimes measured from different devices based on Cs-doped and Ag-doped EILs.

(3) The OLED performance data of the device without EIL should add to evaluate the

effect of the n-doping layer on the OLED performances.

Author reply:

→Thank you for your nice suggestion, the OLED performance data of the device without EIL have been added, as shown from Fig. S4. Without any n-dopant, electrons are hardly injected from cathode, thus leading to rather high driving voltage and low power efficiency.

Fig. S4. The performance of OLED without using any n-dopant for electron injection.

Reviewer 3

(1) What is completely missing however is a proof of the generality of these effects.

The specific material chosen here has in a similar form been used a long time ago for OLED but is not useful for highly stable commercial devices. From the data shown in the paper, it is not possible to conclude the generality of the effect.

→Thank you for your nice question. In order to demonstrate the generality of the doping effect, we use another stable ETM with coordination ability reported by Xiao et al. recently (*Adv. Funct. Mater.* **2018**, 1800429.) and compare the n-doping efficiency with BPhen and TPBi. As shown from Fig. S3(D), except for ETMs of BPhen derivatives, the n-doping method could also be used for other kind of ETMs with strong coordination tendency, such as 2,7-di([2,2':6',2''-terpyridin]-4'-yl)-9,9'-spirobifluorene (27-TPSF). For example, when Ag is doped into 27TPSF, it exhibits a similar high current density in EODs, only a slightly decrease from $\sim 10^2$ to ~ 10 m/A² at a 5 V bias compared with device using BPhen as the ligand. But when Ag is doped into 1,3,5-tris(1-phenyl-1H-benzimidazol-2-yl)benzene (TPBi) with a similar energy level as BPhen but negligible chelating property, no improvement in electron injection can be observed and the current density of EOD keeps at a rather low level (10^{-2} m/A² at a 5 V bias).

Fig. S3 (D). Current density-voltage characteristics of electron-only devices (EODs) with different organic ligands and their molecular structures (ITO/ BPhen (100 nm)/ Ag-doped ligand = 20 % (5 nm)/ Al).

Moreover, although it has been widely known that pristine BPhen is not an ideal electron transport material for a stable device, here with a strong interaction between Ag⁺ and BPhen, the film stability could be largely improved as shown in Fig. 3.

Reviewer 4

(1) The energetics. In Line 90 the authors discussed the huge gain in energy of chelation of Ag. They should piece together the whole energetics, including that of Ag to Ag⁺ and e⁻ to demonstrate the complete chemistry of the process.

→Thank you for your nice suggestion, and we have added the coordination part into the discussion in the manuscript to show the complete chemistry of the ionization process. “But as calculated (Fig. 1A), the ionization energy of Ag is 8.0 eV ($Ag \rightarrow Ag^+ + e^-, 8.0 eV$), but when Ag is doped into BPhen, the ionization energy of Ag decrease rapidly to 4.4 eV ($Ag + BPhen \rightarrow [Ag(BPhen)]^+ + e^-, 4.4 eV$) and 2.7 eV ($Ag + 2BPhen \rightarrow [Ag(BPhen)_2]^+ + e^-, 2.7 eV$) which is even lower than that of cesium, due to the irreversible coordination between Ag⁺ and BPhen and the formation of stable complexes ($Ag^+ + BPhen \rightarrow [Ag(BPhen)]^+, -3.6 eV$ and $[Ag(BPhen)]^+ + BPhen \rightarrow [Ag(BPhen)_2]^+, -1.7 eV$).”

(2) The energy level schematics in Fig. 5B is obtained from Fig. S3B. Fig. S3B should be moved to Fig. 2 to present this important data.

→Thank you for your good suggestion. We have moved Fig. S3B to Fig. 5B followed by the schematic energy-level diagrams to make readers clear about the energy change upon doping.

(3) From the right panel of Fig. S3B, it looks the HOMO position of Ag:p-MeO-Phen is 3.6 eV or higher. They should verify it and mark on the figure how the position of the HOMO is extrapolated, and modify Fig. 5B if necessary.

→Thank you for your good suggestion. The HOMO position is derived from the intersection of the two tangent lines in the low binding energy region. And it mainly depends on the tangents we choose. Here, in order to compare the different level of the HOMO positions, we have to choose the same kind of tangent lines (in the flat

area or in the drop area). With careful consideration, we admit that it is more accurate to use the tangent lines shown in Fig. 5(B) rather than previous used in the flat area, thus we have added the tangent line in Fig. 5(B) to make readers more clear about the method and also revised the schematic energy-level diagrams as follows.

Fig. 5. (B) The UPS spectra and (C) the schematic energy-level diagrams of pristine, Cs-doped and Ag-doped p-MeO-Phen films for comparison.

(4) Another key to being doping instead of forming new compounds is to check if the energy gap has changed. To clarify that, the authors should add IPES data of doped samples.

→Thank you for your nice question and good suggestion. But it needs a special experimental condition to carryout LEIPS measurement and we have measured the LUMO level of p-MeO-Phen using LEIPS with the kind help of Profs. Chihaya Adachi and Toshinori Matsushima in Kyushu University. Here it's even much more difficult to measure the doped samples which requires an *in-situ* film film-fabrication and energy-characterization system. However, except for the LEIPS, we could also measure the absorption spectra of pristine BPhen and Ag-doped BPhen films to check whether the energy gap has changed upon doping. As shown from Fig. S5(C), the absorption spectra of pristine BPhen and Ag-doped BPhen films have basically the same absorption peak with only a slight change in intensity, indicating that the energy gap has been kept at a same level upon doping and the energy change mainly happens in the Fermi level of the film.

Fig. S5. (C) The absorption spectra of pristine and Ag-doped p-MeO-Phen films.

(5) Furthermore, they should compare the shift of the HOMO with those of the core levels, as it is essential in doping that all energy levels of non-reactive elements shift the same amount. This analysis may also address the concern of Reviewer 1 on how to distinguish the doping induced and chemically induced energy level shift.

→Thank you for your nice question. Besides the previous discussion, here we find that the doping induced and chemically induced energy level shift could be also distinguished from nuclear magnetic resonance (NMR), a direct and easy way to study the interaction between molecules, as shown from Fig. S2. Compared with pristine BPhen, the coordination interaction between BPhen and Ag^+ would largely reduce the electron density of 1,10-phenanthroline ring, thus leading to the upward chemical shifts of the hydrogen atoms in 1,10-phenanthroline ring. For example, the chemical shifts are all moved upwards from 9.25, 7.86 and 7.60 ppm to 9.28, 7.90 and 7.72 ppm for H2, H5 and H3, respectively. However, for Ag-doped BPhen, the upward movements of H3 and H5 are only about 0.2-0.3 ppm, which are smaller than Ag^+ -doped one. And as for H2 which locates closer to nitrogen atom, the chemical shift moves in the opposite direction from 9.25 ppm to 9.24 ppm, which means that although there exists a coordination interaction between BPhen and Ag^+ , the strong n-doping effect between Ag and BPhen could greatly increase the electron density in 1,10-phenanthroline ring. Thus it leads to the downward chemical movement of H2 and only a slightly upward chemical movement of H3 and H5.

Fig. S2. The comparison of chemical shifts for pristine BPhen, Ag^+ -doped BPhen and Ag-doped BPhen. (For pristine BPhen (black) and Ag^+ :BPhen (red), their films were firstly prepared by vacuum deposition, following by CDCl_3 dissolving and nuclear magnetic resonance (NMR) measurement. And for Ag^+ +BPhen (blue), AgNO_3 is

added to CDCl₃ solution of BPhen for NMR measurement.)

(6) Inconsistency between the figure and description. In Line 175, it says LEIPS, while in Fig. S3B it is IPES.

→Thank you for your nice comments, and we have revised our manuscript. In Fig. S3B, the wrong expression of IPES has been changed to LEIPS.

(7) The process of the doping should be added to Materials and Methods.

→Thank you for your nice suggestion, and we have added the doping process to Materials and Methods as “Metals and ETMs were co-evaporated inside a high vacuum chamber at 2×10^{-4} Pa. The coating rate of ETM was 1 Å/s, and for metal such as Ag, the coating rate varied from 0.05 to 0.3 Å/s according to the doping concentration.”.

REVIEWERS' COMMENTS:

Reviewer #2 (Remarks to the Author):

The manuscript has been much improved and I recommend that it be accepted for publication.

Reviewer #3 (Remarks to the Author):

In the new version, the authors have added many further measurement data, such as NMR and optical data, which have clarified the open questions. They have also answered my question for generality by showing a new n-material which proves that the effect works also there.

Overall, I believe that the manuscript could be published, however, after another language polish to remove e.g. such glitches like "none-doped" in the figures.

Reviewer #4 (Remarks to the Author):

The authors have adequately addressed the concerns of the reviewers and the manuscript is now publishable in Nat. Comm.